# LEARNING REUSABLE OPTIONS FOR MULTI-TASK REINFORCEMENT LEARNING

## ABSTRACT

Reinforcement learning (RL) has become an increasingly active area of research in recent years. Although there are many algorithms that allow an agent to solve tasks efficiently, they often ignore the possibility that prior experience related to the task at hand might be available. For many practical applications, it might be unfeasible for an agent to learn how to solve a task from scratch, given that it is generally a computationally expensive process; however, prior experience could be leveraged to make these problems tractable in practice. In this paper, we propose a framework for exploiting existing experience by learning reusable *options*. We show that after an agent learns policies for solving a small number of problems, we are able to use the trajectories generated from those policies to learn reusable options that allow an agent to quickly learn how to solve novel and related problems.

## 1 INTRODUCTION

Reinforcement learning (RL) techniques have experienced much of their success in simulated environments, such as video games (Mnih et al., 2015) or board games (Silver et al., 2016; Tesauro, 1995). One of the main reasons why RL has worked so well in these applications is that we are able simulate millions of interactions with the environment in a relatively short period of time. In many real world applications, however, where the agent interacts with the physical world, it might not be easy to generate such a large number of interactions. The time and cost associated with training such systems could render RL an unfeasible approach for training in large scale.

As a concrete example, consider training a large number of humanoid robots (agents) to move quickly, as in the Robocup competition (Farchy et al., 2013). Although the agents have similar dynamics, subtle variations mean that a single policy shared across all agents would not be an effective solution. Furthermore, learning a policy from scratch for each agent is too data-inefficient to be practical. As shown by Farchy et al. (2013), this type of problem can be addressed by leveraging the experience obtained from solving a related task (e.g., walking) to quickly learn a policy for each individual agent that is tailored to a new task (e.g., running). These situations also occurs in industry, such as robots tasked with sorting items in fulfillment centers. A simple approach, like using PD controllers, would fail to adapt to the forces generated from picking up objects with different weight distributions, causing the arm to drop the objects. RL is able to mitigate this problem by learning a policy for each arm that is able to make corrections quickly, which is tailored to the robot's dynamics. However, training a new policy for each agent would be far too costly to be a practical solution. In these scenarios, it is possible to use a small number of policies learned a subset of the agents, and then leverage the experience obtained from learning those policies to allow the remaining agents to quickly learn their corresponding policies. This approach can turn problems that are prohibitively expensive to solve into relatively simple problems.

To make use of prior experience and improve learning on new related problems in RL, several lines of work, which are complementary to each other, have been proposed and are actively being studied. *Transfer learning* (Taylor & Stone, 2009) refers to the problem of adapting information acquired while solving one task to another. One might consider learning a mapping function that allows for a policy learned in *one* task to be used in a different task (Ammar et al., 2015) or simply learn a mapping of the value function learned in one task to another (Taylor et al., 2007). These techniques can be quite effective, but are also limited in that they consider mapping information from *one* source task to *another* target task. Another approach to reusing prior knowledge is through *meta learning*

or learning to learn (Schmidhuber, 1995; Schmidhuber et al., 1998). In the context of RL, the goal under this framework for an agent to be exposed to a number of tasks where it can learn some general behavior that generalizes to new tasks (Finn et al., 2017).

One last technique to leverage prior experience, and the one this paper focuses on, is through temporally extended actions or *temporal abstractions* (McGovern & Sutton, 1998; Sutton et al., 1999). While in the standard RL framework the agent has access to a set of primitive actions (i.e., actions that last for one time step), temporally extended actions allow an agent to execute actions that last for several time-steps. They introduce a bias in the behavior of the agent which, if appropriate for the problem at hand, results in dramatic improvements in how quickly the agent learns to solve a new task. A popular representation for temporally extended actions is the *options* framework (Sutton & Precup, 1998; Sutton et al., 1999) (formally introduced in the next section), which is the focus of this work. It has been shown that options learned in a specific task or set of tasks, can be reused to improve learning on new tasks (Machado et al., 2017; Bacon et al., 2017); however, this often requires knowledge from the user about which options or how many options are appropriate for the type of problems the agent will face.

In this paper, we propose learning reusable options for a set of related tasks with minimal information provided by the user. Throughout this paper, we refer as (near)-optimal policies to those policies that were learned to solve a particular task, but are not strictly speaking optimal. We consider the scenario where the agent must solve a large numbers of tasks and show that after learning a (near)-optimal policy for a small number of problems, we can learn an appropriate number of options that facilitates learning in a remaining set of tasks. To do so, we propose learning a set of options that minimize the expected number of decisions needed to represent trajectories generated from the (near)-optimal policies learned by the agent, while also maximizing the probability of generating those trajectories. Unlike techniques that learn options to rach bottleneck states (McGovern & Barto, 2001) or states deemed of high value (Machado et al., 2017), our method seeks to learn options that are able to generate trajectories known to perform well. This does not necessarily lead to learn options that reach states one might consider "interesting".

## 2 BACKGROUND AND NOTATION

A *Markov decision process* (MDP) is a tuple, $M = (\mathcal{S}, \mathcal{A}, P, R, \gamma, d_0)$, where $\mathcal{S}$ is the set of possible states of the environment, $\mathcal{A}$ is the set of possible actions that the agent can take, $P(s, a, s')$ is the probability that the environment will transition to state $s' \in \mathcal{S}$ if the agent executes action $a \in \mathcal{A}$ in state $s \in \mathcal{S}$, $R(s, a, s')$ is the expected reward received after taking action $a$ in state $s$ and transitioning to state $s'$, $d_0$ is the initial state distribution, and $\gamma \in [0, 1]$ is a discount factor for rewards received in the future. We use $t$ to index the time-step and write $S_t$, $A_t$, and $R_t$ to denote the state, action, and reward at time $t$. A *policy*, $\pi : \mathcal{S} \times \mathcal{A} \to [0, 1]$, provides a conditional distribution over actions given each possible state: $\pi(s, a) = \Pr(A_t = a | S_t = s)$. We denote a trajectory of length $t$ as $h_t = (s_0, a_0, r_0, \ldots, s_{t-1}, a_{t-1}, r_{t-1}, s_t)$, that is, $h_t$ is defined as a sequence of states, actions and rewards observed after following some policy for $t$ time-steps. This work focuses on learning options that can be used for a set of related tasks. We consider the setting where an agent must solve a set of related tasks, where each task is an MDP, $M = (\mathcal{S}, \mathcal{A}, P_M, R_M, \gamma, d_0^M)$; that is, each task is an MDP with its own transition function, reward function and initial state distribution, with shared state and action sets.

An option, $o = (\mathcal{I}_o, \mu_o, \beta_o)$, is a tuple in which $\mathcal{I}_o \subseteq \mathcal{S}$ is the set of states in which option $o$ can be executed (the *initiation set*), $\mu_o$ is a policy that governs the behavior of the agent while executing $o$, and $\beta_o : \mathcal{S} \to [0, 1]$ is a termination function that determines the probability that $o$ terminates in a given state. We assume that $\mathcal{I}_o = \mathcal{S}$ for all options $o$; that is, the options are available at every state. The options framework does not dictate how an agent should choose between available options or how options should be discovered. A common approach to selecting between options is to a learn a *policy over options*, which is defined by the probability of choosing an option in a particular state. Two recent popular approaches to option discovery are eigenoptions (Machado et al., 2017) and the option-critic architecture (Bacon et al., 2017).

The *eigenoptions* (Machado et al., 2017) of an MDP are the optimal policies for a set of implicitly defined reward functions called *eigenpurposes*. Eigenpurposes are defined in terms of *proto-value functions* (Mahadevan, 2005), which are in turn derived from the eigenvectors of a modified adjacency matrix over states for the MDP. The intuition is that no matter the true reward function, the

eigenoptions allow an agent to quickly traverse the transition graph, resulting in better exploration of the state space and faster learning. However, there are two major downsides: 1) the adjacency matrix is often not known *a priori*, and may be difficult to construct for large MDPs, and 2) for each eigenpurpose, constructing the corresponding eigenoption requires solving a new MDP. The option-critic architecture (Bacon et al., 2017) is a more direct approach to learn options and a policy over options simultaneously using policy gradient methods. One issue that often arises within this framework is that the termination functions of the learned options tend to collapse to "always terminate". In a later publication, the authors built on this work to consider the case where there is a cost associated with switching options (Harb et al., 2018). This method resulted in the agent learning to use a single option while it was appropriate and terminate when an option switch was needed, allowing it to discover improved policies for a particular task. The authors argue that minimizing the use of the policy over options may be desirable, as the cost of choosing an option may be greater than the cost of choosing a primitive action when using an option. Recent work by Harutyunyan et al. (2019) approaches the aforementioned termination problem by explicitly optimizing the termination function of options to focus on small regions of the state space. However, in contrast to the work presented in these paper, these methods do not explicitly take into consideration that the agent might face many related tasks in the future.

We build on the idea that minimizing the number of decisions made by an agent leads to the discovery of general reusable options, and propose an offline method where they are learned by solving a small number of tasks. The options are then leveraged to quickly solve new problems the agent will face in the future. We use the trajectories generated while learning (near)-optimal policies, and learn an appropriate set of options by directly minimizing the expected number of decisions the agent makes while simultaneously maximizing the probability of generating the observed trajectories.

## 3 LEARNING REUSABLE OPTIONS FROM EXPERIENCE

In this section, we introduce the objective for learning a set of reusable options for a set of related tasks. Our algorithm introduces one option at a time until introducing a new option does not improve the objective further. This procedure results in a natural way of learning an adequate number of options without having to predefine it; a new option is included if it is able to improve the probability of generating optimal behavior while minimizing the number of decisions made by the agent. Our method assumes that the agent has learn a policy for a small number of tasks, and sample trajectories are obtained from these (near)-optimal policies. Notice that the propose algorithm is only concerned with being able to recreate the demonstrated trajectories, so if these were sample from a poorly performing policy the options learned are unlikely to provide any benefits.

### 3.1 PROBLEM FORMULATION

In the options framework, at each time-step, $t$, the agent chooses an action, $A_t$, based on the current option, $O_t$. Let $T_t$ be a Bernoulli random variable, where $T_t = 1$ if the previous option, $O_{t-1}$, terminated at time $t$, and $T_t = 0$ otherwise. If $T_t = 1$, $O_t$ is chosen using the policy over options, $\pi$. If $T_t = 0$, then the previous option continues, that is, $O_t = O_{t-1}$. To ensure we can represent any trajectory, we consider primitive actions to be options which always select one specific action and then terminate; that is, for an option, $o$, corresponding to a primitive, $a$, for all $s \in \mathcal{S}$, the termination function would be given by $\beta_o(s) = 1$, and the policy by $\mu(s, a') = 1$ if $a' = a$ and 0 otherwise.

Let $\mathcal{O} = \mathcal{O}_\mathcal{A} \cup \mathcal{O}_\mathcal{O}$ denote a set of options, $\{o_1, \ldots, o_n\}$, where $\mathcal{O}_\mathcal{A}$ refers to the set of options corresponding to primitive actions and $\mathcal{O}_\mathcal{O}$ to the set corresponding to temporal abstractions. Furthermore, let $H$ be a random variable denoting a trajectory of length $|H|$ generated by a near-optimal policy, and let $H_t$ be a random variable denoting the sub-trajectory of $H$ up to the state encountered at time-step $t$. We seek to find a set, $\mathcal{O}^* = \{o_1^*, \ldots, o_n^*\}$, that maximizes the following objective:

$$J(\pi, \mathcal{O}) = \mathbf{E}\Big[\sum_{t=1}^{|H|} \Pr(T_t = 0, H_t | \pi, \mathcal{O}) + \lambda_1 g(H, \mathcal{O}_\mathcal{O})\Big], \tag{1}$$

where $g(h, \mathcal{O}_\mathcal{O})$ is a regularizer that encourages a diverse set of options, and $\lambda_1$ is a scalar hyper-parameter. If we are also free to learn the parameters of $\pi$, then $\mathcal{O}^* \in \arg\max_\mathcal{O} \max_\pi J(\pi, \mathcal{O})$.

One choice for $g$ is the average KL divergence on a given trajectory over the set of $m$ options

being learned: $g(h, \mathcal{O}_\mathcal{O}) = \frac{2}{m(m-1)} \sum_{o,o' \in \mathcal{O}_\mathcal{O}} \sum_{t=0}^{|h|-1} D_{\mathrm{KL}}\left(\mu_o(s_t) \| \mu_{o'}(s_t)\right).$ [1] Intuitively, we seek to find options that are capable of generating near-optimal trajectories with a small number of terminations. Notice that minimizing the number of terminations is the same as minimizing the number of decisions made by the policy over options, as each termination requires the policy to choose a new option. Given a set of options, a policy over options, and a near-optimal sample trajectory, we can calculate the joint probability for a trajectory *exactly*, and estimate equation 1 by averaging over a *set* of near-optimal trajectories.

## 3.2 OPTIMIZATION OBJECTIVE FOR LEARNING OPTIONS

Given that the agent must solve a set of tasks, we can use the experienced gathered on a subset of tasks to obtain trajectories demonstrating optimal behavior. Given a set, $\mathcal{H}$, of trajectories generated from an initial subset of tasks, we can now estimate the expectation in equation 1 to learn options that can be leveraged in the remaining problems. Because the probability of generating any trajectory approaches 0 as the length of the trajectory increases, we make modify the original objective for better numerical stability, and arrive to the objective $\hat{J}$ that we optimize in practice.

$$\hat{J}(\pi, \mathcal{O}, \mathcal{H}) = \frac{1}{\mathcal{H}} \sum_{h \in \mathcal{H}} \Big( \underbrace{\lambda_2 \Pr(H = h | \pi, \mathcal{O})}_{\text{probability of generating } h} - \underbrace{\frac{\sum_{t=1}^{|h|} \mathbf{E}\left[T_t = 1 | H_t = h_t, \pi, \mathcal{O}\right]}{|h|}}_{\text{expected number of terminations}} + \underbrace{\lambda_1 g(h, \mathcal{O}_\mathcal{O})}_{\text{encourage diverse options}} \Big) \quad .$$

(2)

A more detailed discussion on how we arrived to this objective is provided in Appendix A. We can express equation 2 entirely in terms of the policy over options $\pi$, options $\mathcal{O} = \{o_1, \ldots, o_n\}$ and the transition function, $P$ (which we estimate from samples). The following theorems show how to calculate the first two terms in equation 2, allowing us to maximize the proposed objective.

**Theorem 1.** *Given a set of options, $\mathcal{O}$, and a policy, $\pi$, over options, the expected number of terminations for a trajectory $h$ is given by:*

$$\sum_{t=1}^{|h|} \mathbf{E}\left[T_t = 1 \Big| H_t = h_t, \pi, \mathcal{O}\right] = \sum_{t=1}^{|h|} \sum_{o \in \mathcal{O}} \beta_o(s_t) \frac{\mu_o(s_{t-1}, a_{t-1}) \Pr(O_{t-1} = o | H_{t-1} = h_{t-1}, \pi, \mathcal{O})}{\sum_{o' \in \mathcal{O}} \mu_o(s_{t-1}, a_{t-1}) \Pr(O_{t-1} = o' | H_{t-1} = h_{t-1}, \pi, \mathcal{O})},$$

(3)

$$\Pr(O_t = o | H_t = h_t, \pi, \mathcal{O}) = \Big[ \Big( \pi(s_t, o)\beta_o(s_t) \Big) + \Big( P(s_{t-1}, a_{t-1}, s_t)\alpha_{t-1}(o)(1 - \beta_o(s_{t-1})) \Big) \Big],$$

*and* $\Pr(O_0 = o | H_0 = h_0, \pi, \mathcal{O}) = \pi(s_0, o)$.

*Proof.* See Appendix B. $\qquad\square$

**Theorem 2.** *Given a set of options $\mathcal{O}$ and a policy $\pi$ over options, the probability of generating a trajectory $h$ of length $|h|$ is given by:*

$$\Pr(H_{|h|} = h_{|h|} | \pi, \mathcal{O}) = d_0(s_0) \Big[ \sum_{o \in \mathcal{O}} \pi(s_0, o)\mu_o(s_0, a_0) f(h_{|h|}, o, 1) \Big] \prod_{k=0}^{|h|-1} P(s_k, a_k, s_{k+1}),$$

*where $f$ is a recursive function defined as:*

$$f(h_t, o, i) = \begin{cases} 1, & \text{if } i = t \\ \Big[ \Big( \beta_o(s_i) \sum_{o' \in \mathcal{O}} \pi(s_{i+1}, o')\mu_{o'}(s_{i+1}, a_{i+1}) f(h_t, o', i+1) \Big) \\ \quad + \Big( (1 - \beta_o(s_i))\mu_o(s_{i+1}, a_{i+1}) f(h_t, o, i+1) \Big) \Big] & \text{otherwise} \end{cases}$$

*Proof.* See Appendix C. $\qquad\square$

---

[1]This term is only defined when we consider more than one option. Otherwise, we set this term to 0.

Given a parametric representation of the option policies and termination functions for each $o \in \mathcal{O}$ and for the policy $\pi$ over options, we use Theorems 1 and 2 to differentiate the objective in equation 2 with respect to their parameters and optimize with any numerical optimization technique.

### 3.3 LEARNING OPTIONS INCREMENTALLY

One common issue in option discovery is identifying how many options are needed for a given problem. Oftentimes this number is predefined by the user based on intuition. In such a scenario, one could learn options by simply randomly initializing the parameters of a number of options and optimizing the proposed objective in equation 2. Instead, we propose not only learning options, but also the number of options needed, by the procedure shown in Algorithm 1. This algorithm introduces one option at a time and optimizes the objective $\hat{J}$ with respect to the policy over options $\pi_\theta$, with parameters $\theta$, and the newly introduced option, $o' = (\mu'_\phi, \beta'_\psi)$, with parameters $\phi$ and $\psi$, for $N$ epochs. Optimizing both $o'$ and $\pi_\theta$ allows us to estimate how much we can improve $\hat{J}$ given

---

**Algorithm 1** Option Learning Framework - Pseudocode

1: Collect set of trajectories $\mathcal{H}$
2: Initialize option set $\mathcal{O}$ with primitive options
3: done = false
4: $\hat{J}_{prev} = -\infty$
5: **while** done == false **do**
6:     Initialize new option $o' = (\mu'_\phi, \beta'_\psi)$, initializing parameters for $\phi$ and $\psi$.
7:     $\mathcal{O}' = \mathcal{O} \cup o'$
8:     Initialize parameters $\theta$ of policy $\pi_\theta$
9:     **for** k=1,...,N **do**
10:         $\hat{J}_k = \hat{J}(\pi_\theta, \mathcal{O}', \mathcal{H})$
11:         $\theta = \theta + \alpha \frac{\partial \hat{J}_k}{\partial \theta}$
12:         $\phi = \phi + \alpha \frac{\partial \hat{J}_k}{\partial \phi}$
13:         $\psi = \psi + \alpha \frac{\partial \hat{J}_k}{\partial \psi}$
14:     **if** $\hat{J}_N - \hat{J}_{prev} < \Delta$ **then**
15:         done = true
16:     **else**
17:         $\mathcal{O} = \mathcal{O}'$
18:         $\hat{J}_{prev} = \hat{J}_N$
19: Return new option set $\mathcal{O}$

---

that we keep any previously introduced option fixed. After the new option is trained, we measure how much $\hat{J}$ has improved; if it fails to improve above some threshold, $\Delta$, the procedure terminates. This results in a natural way of obtaining an appropriate number of options, as options stop being added once a new option no longer improves the ability to represent the demonstrated behavior.

## 4 EXPERIMENTAL RESULTS

This section describes experiments used to evaluate the proposed technique approach. We show results in the "four rooms" domain to allow us to visualize and understand the options produced by our method, and to show empirically that these options produce a clear improvement in learning. We use this domain to show that options generated by our method are able to generalize to tasks where the option-critic architecture (Bacon et al., 2017) and eigenoptions (Machado et al., 2017) would fail to do so. We then extend our experiments to evaluate our technique in a few selected problems from the Atari 2600 emulator provided by OpenAI Gym (Brockman et al., 2016). These experiments demonstrate that by using the trajectories obtained from solving a small subset of tasks, our approach is able to discover options that significantly improve the learning ability of the agent in the tasks it has yet to solve. For the four room experiment, we assume the transition function was known in advance. In all ATARI experiments, we estimated the transition functions by fitting the parameters of a linear Gaussian model to all the transitions experienced during training.

### 4.1 EXPERIMENTS ON FOUR ROOMS ENVIRONMENT

We tested our approach in the four rooms domain: a gridworld of size $40 \times 40$, in which the agent is placed in a start state and needs to reach a goal state. At each time-step, the agent executes one of four available actions: moving left, right, up or down, and receives a reward of $-1$. Upon reaching the goal state, the agent receives a reward of $+10$. We generated 30 different task variations by changing the goal and start locations, and collected six sample trajectories from optimal policies learned in six tasks. We evaluated our method on the remaining 24 tasks.

Figure 1a shows the change in the average expected number of terminations and average probability of generating the observed trajectories while learning options, as new options are introduced and adapted to the sampled trajectories. Options were learned over the six sampled optimal trajectories and every 50 epochs a new option was introduced. For every new option, the change in probability of

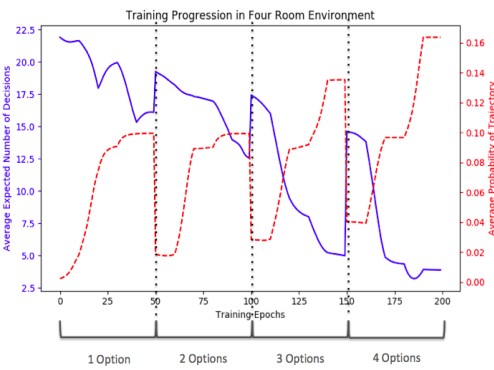
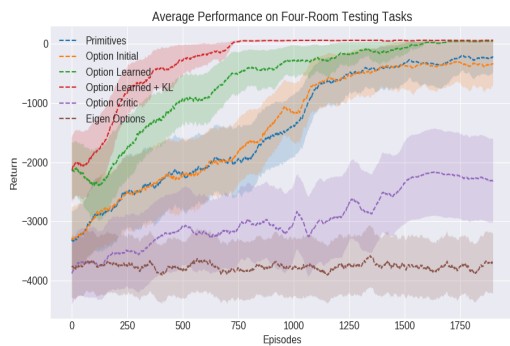

(a) Visualization of loss over 200 training epochs for the four rooms domain. The decreasing average number of decisions made by $\pi$ is shown in blue and the increasing probability of generating the sampled trajectories is shown in red.

(b) Performance comparison on four rooms domain. Six tasks were used for training and 24 different for testing. The plot shows the average return (and standard error) on the y-axis as a function of the episode number on the test tasks.

Figure 1: Results on four-room domain. Six tasks were used for training and 24 for testing.

generating the observed trajectories as well as the change in expected number of decisions reaches a plateau after 30 or 40 training epochs. When a new option is introduced, there is a large jump in the loss because a new policy, $\pi$, is initialized arbitrarily to account for the new option set being evaluated. However, after training the new candidate option, the overall loss improves beyond what it was possible before introducing the new option.

In Figure 1b, we compare the performance of Q-learning on 24 novel test tasks using options discovered by our method (with and without regularization using KL divergence), eigenoptions, and option critic. We allowed each competing method to learn options from the same six training tasks and, to ensure a fair comparison, we used the original code provided by the authors. As baselines, we also compare against primitive actions and randomly initialized options. It might seem surprising that both eigenoptions and the option-critic failed to reach an optimal policy when they were shown to work well in this type of problem; for that we offer the following explanation. Our implementation of four rooms is defined in a much larger state space than the ones where these methods were originally tested, making each individual room much larger. Since the options identified by these methods tend to lead the agent from room to room, it is possible that, once in the correct room, the agent executes an option leading to a different room before it had the opportunity to find the goal. When testing our approach in the smaller version of the four room problem, we found no clear difference in performance of the competing methods. In this experiment, we set the threshold $\Delta$ for introducing a new option to $10\%$ of $\hat{J}$ at the previous iteration and the hyperparameter $\lambda_2 = 100.0$. When adding KL regularization, we set $\lambda_1 = 0.001$.

Figure 2 shows a visualization of the policy learned by the agent for a specific task. The policy leads the agent to navigate from a specific location in the bottom-left room to a location in the top-right room in a small "four-room" domain of size $10 \times 15$. [2] The new task to solve is shown in the top-left figure, while the solution found is shown in the top-right figure. The remaining rows of images depict the learned option policies, termination functions, and how they were used in the new task. The first row shows the learned option policies after training, the center row depict the termination functions and the bottom row shows a heat-map depicting where each option is likely to be called. The figure shows that while the options are defined over the entire state space, they are only useful in specific regions—that is, they are specialized. These options, when used in combination in specific regions, allow the agent to learn how to solve new problems more efficiently.

## 4.2 Experiments using Atari 2600 Games

We evaluated the quality of the options learned by our framework in two different Atari 2600 games: Breakout and Amidar. We trained the policy over options using A3C (Mnih et al., 2016) with grayscale pixel input. Options were represented by a two layer convolutional neural network, and

---

[2]We show a smaller domain than used in the experiments for ease of visualization

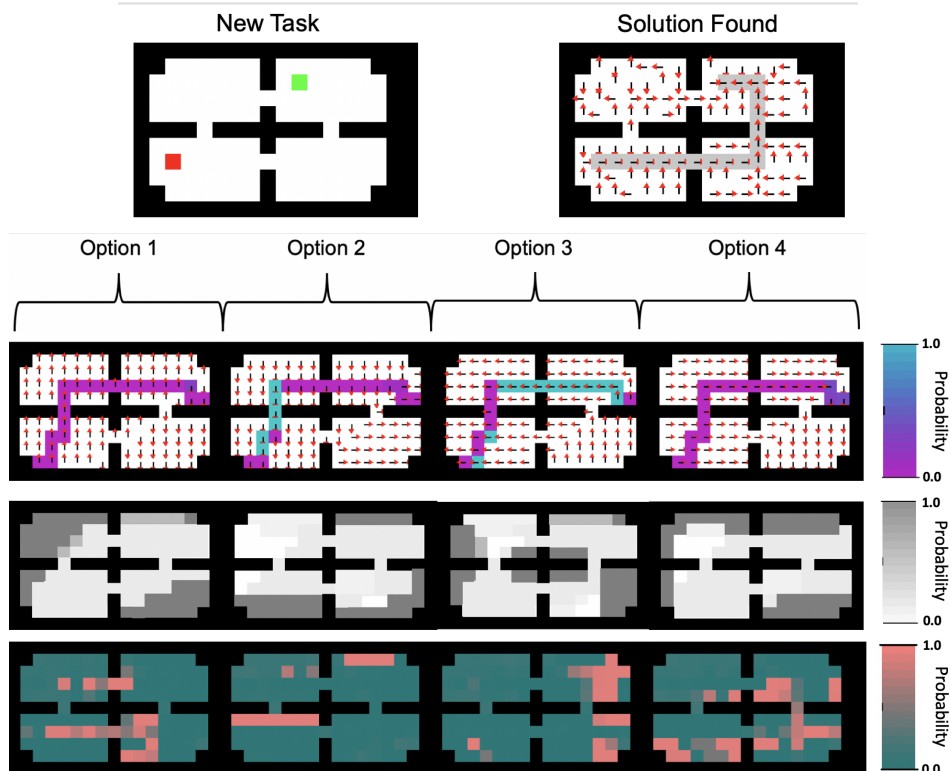

Figure 2: Visualization of our framework in four rooms domain. A novel task is seen in the top left, where the agent (red) has to navigate to a goal (green). On the top right, we show the solution found by the agent. The three rows below show how the options were learned and exploited in the new task. The highlighted area in the top row shows a sample trajectory and the color corresponds to the probability that the option would take the demonstrated action. Notice that this trajectory was obtained on a previous tasks, so it does not correspond to the new task on top. The arrows show the action that is most likely at each state. After training (first row) each option specializes in a specific skill (a navigation pattern). In this case, the demonstrated trajectory can be generated by using option 3 and 2. The middle rows shows a heat-map indicating where an option is likely to terminate (close to walls), and the last row shows where each options is likely to be used by the policy learned in the new task. The agent learns to use each option in specific situations; for example, option 1 is likely to be called to make the agent move up, if it is located in one of the bottom rooms.

were given the previous two frames as input. In both experiments the task variations consisted in changing the number of frames skipped after taking an action (randomly selected between 2 and 10), the reward function by scaling the reward with a real number between 0.1 and 10.0, and initial state distribution by letting the agent execute between 0 and 20 actions before start it starts learning. The full implementation details for these experiments are given in Appendix E. Figures 3a and 3b show the performance of the agent as a function of training time in Breakout and Amidar, respectively. The plots show that given good choices of hyperparameters, the learned options led to a clear improvement in performance during training. For both domains, we found that $\lambda_2 = 5,000$ led to a reasonable trade-off between the first two term in $\hat{J}$, and report results with three different regularization values: $\lambda_1 = 0.0,$, $\lambda_1 = 0.01$ and $\lambda_1 = 0.1$.

Note that our results do not necessarily show that the options result in a better final policy, but they improve exploration early in training and enable the agent to learn more effectively. Figure 4a depicts the behavior for one of the learned options on Breakout. The option efficiently catches the ball after it bounces off the left wall, and then terminates with high probability before the ball has to be caught again. Bear in mind that the option remains active for many time-steps, significantly reducing the number of decisions made by the policy over options. However, it does not maintain control for so long that the agent is unable to respond to changing circumstances. Note that the option is only useful in specific case; for example, it was not helpful in returning a ball bounced off the right wall. That is to say, the option specialized in a specific sub-task within the larger problem:

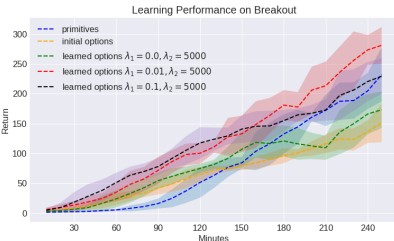
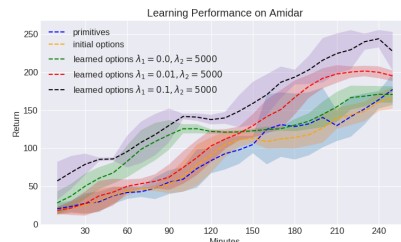

(a) Average returns on novel tasks for Breakout.

(b) Average returns on novel tasks for Amidar.

Figure 3: Comparison on Atari domains for primitives (blue), options before training (orange) and learned options for different values of $\lambda_1$ and $\lambda_2$. Shaded regions indicate standard error.

a highly desirable property for generally useful options. Figure 4b shows the selection of two of the options learned for Amidar when starting a new game. At the beginning of the game, option 1 is selected, which takes the agent to a specific intersection before terminating. The agent then selects option 2, which chooses a direction at the intersection, follows the resulting path, and terminates at the next intersection. Note that the agent does not need to repeatedly select primitive actions in order to simply follow a previously chosen path. Having access to these types of options enables an agent to easily replicate known good behaviors, allowing for faster and more meaningful exploration of the state space.

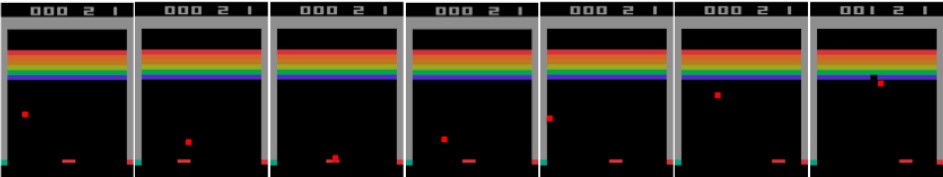

(a) Visualization of a learned option executed until termination on Breakout. The option learned to catch the ball bouncing off the left wall and terminates with high probability before the ball bounces a wall again (ball size increased for visualization).

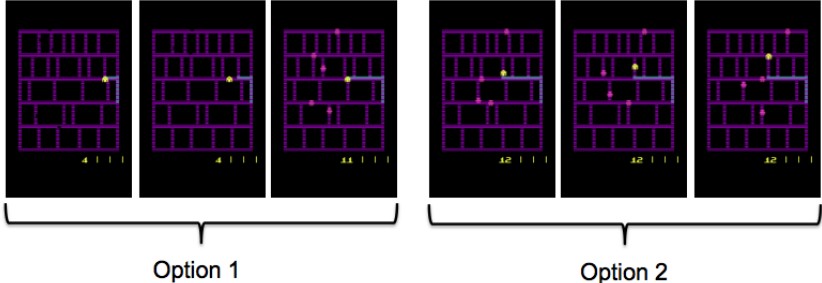

(b) Visualization of two learned options on Amidar. The agent is shown in yellow and enemies in pink. Option 1 learned to move up, at the beginning of the game, and turn left until reaching an intersection. Option 2 learned to turn in that intersection and move up until reaching the next one.

## 5 CONCLUSION AND FUTURE WORK

In this work we presented an optimization objective for learning options offline from demonstrations of near-optimal behavior on a set of tasks. Optimizing the objective results in a set of options that allows an agent to reproduce the behavior while minimizing the number of decisions made by the policy over options, which are able to improve the learning ability of the agent on new tasks. We provided results showing how options adapt to the trajectories provided and showed, through several experiments, that the identified options are capable of significantly improving the learning ability of an agent. The resulting options encode meaningful abstractions that help the agent interact with and learn from its environment more efficiently.

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

## A  APPENDIX

The following list defines the notation used in all derivations:

1. $A_t$: random variable denoting action taken at step $t$.
2. $S_t$: random variable denoting state at step $t$.
3. $H_t$: random variable denoting history up to step $t$. $H_t = (S_0, A_0, S_1, A_1, \ldots, S_t)$.
4. $T_t$: random variable denoting the event that the option used at step $t-1$ terminates at state $S_t$.
5. $\pi$: policy over options.
6. $P$: transition function. $P(s, a, s')$ denotes the probability of transitioning to state $s'$ by taking action $a$ in state $s$
7. $O_t$: random variable denoting the option selected for execution at state $S_t$.
8. $o$: option defined as $o = (\mu_o, \beta_o)$, where $\mu_o$ is the option policy for option and $\beta_o$ is the termination function.
9. Assume primitives are options that perform only 1 action and last for 1 time-step.
10. $\mathcal{O}$: set of available options.

We can compute the probability of an option terminating at state $s_t$ and generating a trajectory $h_t$ as:

$$\Pr(T_t = 1, H_t = h_t | \pi, \mathcal{O}) = \Pr(T_t = 1 | H_t = h_t, \pi, \mathcal{O}) \Pr(H_t = h_t | \pi, \mathcal{O}) \qquad (4)$$

To compute the proposed objective $J$ we need to find an expression for $\Pr(T_t = 1 | H_t = h_t, \pi, \mathcal{O})$ and $\Pr(H_t = h_t | \pi, \mathcal{O})$ in terms of known quantities.

### A.1  APPENDIX A - DERIVATION OF $\hat{J}$

Recall $J(\pi, \mathcal{O}, H) = \mathbf{E}\left[\sum_{t=1}^{|h|} \Pr(T_t = 0, H_t | \pi, \mathcal{O})\right]$, ignoring the regularization term. Assuming access to a set $\mathcal{H}$ of sample trajectories, we start by estimating $J$ from sample averages and derive the objective $\hat{J}$ as follows:

$$J(\pi, \mathcal{O}, \mathcal{H}) \approx \frac{1}{|\mathcal{H}|} \sum_{h \in \mathcal{H}} \sum_{t=1}^{|h|} \Pr(T_t = 0, H_t = h_t | \pi, \mathcal{O})$$

$$= \frac{1}{|\mathcal{H}|} \sum_{h \in \mathcal{H}} \sum_{t=1}^{|h|} \left(1 - \Pr(T_t = 1 | H_t = h_t, \pi, \mathcal{O})\right) \Pr(H_t = h_t | \pi, \mathcal{O})$$

$$= \frac{1}{|\mathcal{H}|} \sum_{h \in \mathcal{H}} \sum_{t=1}^{|h|} \left(1 - \mathbf{E}\left[T_t | H_t = h_t, \pi, \mathcal{O}\right]\right) \Pr(H_t = h_t | \pi, \mathcal{O})$$

It can easily be seen that to maximize the above expression $\mathbf{E}\left[T_t | H_t = h_t, \pi, \mathcal{O}\right]$ should be minimized while $\Pr(H = h | \pi, \mathcal{O})$ should be maximized. Given that for long trajectories the expected number of terminations increases while the probability of generating the trajectories goes to 0, we normalize the number of terminations by the lenght of the trajectory, $|h|$, and adjust a hyperparameter, $\lambda_2$, to prevent one term from dominating the other during optimization. Based on this observation we propose optimizing the following objective:

$$\hat{J}(\pi, \mathcal{O}, \mathcal{H}) = \frac{1}{\mathcal{H}} \sum_{h_{|h|} \in \mathcal{H}} \lambda_2 \Pr(H = h | \pi, \mathcal{O}) - \frac{\sum_{t=1}^{|h|} \mathbf{E}\left[T_t | H_t = h_t, \pi, \mathcal{O}\right]}{|h|}.$$

This objective allow us to control a trade-off, through $\lambda_2$, of how much we care about the options reproducing the demonstrated trajectories vs. how much we want the agent to minimize the number of decisions.

## A.2  APPENDIX B - PROOF OF THEOREM 1

**Theorem 1** *Given a set of options $\mathcal{O}$ and a policy $\pi$ over options, the expected number of terminations for a trajectory $h$ of length $|h|$ is given by:*

$$\sum_{t=1}^{|h|} \mathbf{E}\left[T_t = 1 | H_t = h_t, \pi, \mathcal{O}\right] = \sum_{t=1}^{|h|} \left( \sum_{o \in \mathcal{O}} \beta_o(s_t) \frac{\mu_o(s_{t-1}, a_{t-1}) \Pr(O_{t-1} = o | H_{t-1} = h_{t-1}, \pi, \mathcal{O})}{\sum_{o' \in \mathcal{O}} \mu_{o'}(s_{t-1}, a_{t-1}) \Pr(O_{t-1} = o' | H_{t-1} = h_{t-1}, \pi, \mathcal{O})} \right),$$

*where,*

$$\Pr(O_{t-1} = o | H_{t-1} = h_{t-1}, \pi, \mathcal{O}) = \left[ \left( \pi(s_{t-1}, o) \beta_o(s_{t-1}) \right) \left( P(s_{t-2}, a_{t-2}, s_{t-1}) \mu_o(s_{t-2}, a_{t-2}) \right. \right.$$
$$\left. \left. \times \Pr(O_{t-2} = o | H_{t-2} = h_{t-2}, \pi, \mathcal{O})(1 - \beta_o(s_{t-1})) \right) \right],$$

*and $\Pr(O_0 = o | H_0 = h_0, \pi, \mathcal{O}) = \pi(s_0, o)$.*

*Proof.* Notice that $\sum_{t=1}^{|h|} \mathbf{E}\left[T_t = 1 | H_t = h_t, \pi, \mathcal{O}\right] = \sum_{t=1}^{|h|} \Pr(T_t = 1 | H_t = h_t, \pi, \mathcal{O})$ 1, so if we find an expression for $\Pr(T_t = 1 | H_t = h_t, \pi, \mathcal{O})$, we can calculate the expectation exactly. We define $\Pr(T_0 = 1 | H_1 = h_1, \pi, \mathcal{O}) = 1$ for ease of derivation even though there is no option to terminate at $T_0$.

$$\Pr(T_t = 1 | H_t = h_t, \pi, \mathcal{O}) = \sum_{o \in \mathcal{O}} \Pr(T_t = 1 | O_{t-1} = o, H_t = h_t, \pi, \mathcal{O}) \Pr(O_{t-1} = o | H_t = h_t, \pi, \mathcal{O})$$

$$= \sum_{o \in \mathcal{O}} \beta_o(s_t) \Pr(O_{t-1} = o | H_t = h_t, \pi, \mathcal{O})$$

$$= \sum_{o \in \mathcal{O}} \beta_o(s_t) \Pr(O_{t-1} = o | H_{t-1} = h_{t-1}, A_{t-1} = a_{t-1}, S_t = s_t, \pi, \mathcal{O})$$

$$= \sum_{o \in \mathcal{O}} \beta_o(s_t) \frac{\Pr(S_t = s_t | H_{t-1} = h_{t-1}, A_{t-1} = a_{t-1}, O_{t-1} = o, \pi, \mathcal{O})}{\Pr(S_t = s_t | H_{t-1} = h_{t-1}, A_{t-1} = a_{t-1}, \pi, \mathcal{O})}$$
$$\times \Pr(O_{t-1} = o | H_{t-1} = h_{t-1}, A_{t-1} = a_{t-1}, \pi, \mathcal{O})$$

$$= \sum_{o \in \mathcal{O}} \beta_o(s_t) \frac{\Pr(S_t = s_t | H_{t-1} = h_{t-1}, A_{t-1} = a_{t-1}, \pi, \mathcal{O})}{\Pr(S_t = s_t | H_{t-1} = h_{t-1}, A_{t-1} = a_{t-1}, \pi, \mathcal{O})}$$
$$\times \Pr(O_{t-1} = o | H_{t-1} = h_{t-1}, A_{t-1} = a_{t-1}, \pi, \mathcal{O})$$

$$= \sum_{o \in \mathcal{O}} \beta_o(s_t) \Pr(O_{t-1} = o | H_{t-1} = h_{t-1}, A_{t-1} = a_{t-1}, \pi, \mathcal{O})$$

$$= \sum_{o \in \mathcal{O}} \beta_o(s_t) \frac{\Pr(A_{t-1} = a_{t-1} | H_{t-1} = h_{t-1}, O_{t-1} = o, \pi, \mathcal{O}) \Pr(O_{t-1} = o | H_{t-1} = h_{t-1}, \pi, \mathcal{O})}{\Pr(A_{t-1} = a_{t-1} | H_{t-1} = h_{t-1}, \pi, \mathcal{O})}$$

$$= \sum_{o \in \mathcal{O}} \beta_o(s_t) \frac{\mu_o(s_{t-1}, a_{t-1}) \Pr(O_{t-1} = o | H_{t-1} = h_{t-1}, \pi, \mathcal{O})}{\Pr(A_{t-1} = a_{t-1} | H_{t-1} = h_{t-1}, \pi, \mathcal{O})}$$

$$
\begin{aligned}
&= \sum_{o \in \mathcal{O}} \beta_o(s_t) \frac{\mu_o(s_{t-1}, a_{t-1}) \Pr(O_{-1} = o | H_{t-1} = h_{t-1}, \pi, \mathcal{O})}{\sum_{o' \in \mathcal{O}} \Pr(A_{t-1} = a_{t-1}, O_{-1} = o' | H_{t-1} = h_{t-1}, \pi, \mathcal{O})} \\
&= \sum_{o \in \mathcal{O}} \beta_o(s_t) \mu_o(s_{t-1}, a_{t-1}) \Pr(O_{-1} = o | H_{t-1} = h_{t-1}, \pi, \mathcal{O}) \\
&\quad \times \Big( \sum_{o' \in \mathcal{O}} \Pr(A_{t-1} = a_{t-1} | O_{-1} = o', H_{t-1} = h_{t-1}, \pi, \mathcal{O}) \\
&\quad \times \Pr(O_{-1} = o' | H_{t-1} = h_{t-1}, \pi, \mathcal{O}) \Big)^{-1} \\
&= \sum_{o \in \mathcal{O}} \beta_o(s_t) \frac{\mu_o(s_{t-1}, a_{t-1}) \Pr(O_{-1} = o | H_{t-1} = h_{t-1}, \pi, \mathcal{O})}{\sum_{o' \in \mathcal{O}} \mu_{o'}(s_{t-1}, a_{t-1}) \Pr(O_{-1} = o' | H_{t-1} = h_{t-1}, \pi, \mathcal{O})}
\end{aligned}
$$

We are left with finding an expression in terms of known probabilities for $\Pr(O_{-1} = o | H_{t-1} = h_{t-1}, \pi, \mathcal{O})$.

$$
\begin{aligned}
\Pr(O_{-1} = o | H_{t-1} = h_{t-1}, \pi, \mathcal{O}) =& \big[ \Pr(O_{-1} = o, T_{t-1} = 1 | H_{t-1} = h_{t-1}, \pi, \mathcal{O}) \\
&+ \Pr(O_{-1} = o, T_{t-1} = 0 | H_{t-1} = h_{t-1}, \pi, \mathcal{O}) \big] \\
=& \Big[ \big( \Pr(O_{-1} = o | H_{t-1} = h_{t-1}, T_{t-1} = 1, \pi, \mathcal{O}) \\
&\times \Pr(T_{t-1} = 1 | H_{t-1} = h_{t-1}, \pi, \mathcal{O}) \big) \\
&+ \big( \Pr(O_{-1} = o | H_{t-1} = h_{t-1}, T_{t-1} = 0, \pi, \mathcal{O}) \\
&\times (1 - \Pr(T_{t-1} = 1 | H_{t-1} = h_{t-1}, \pi, \mathcal{O})) \big) \Big] \\
=& \Big[ \big( \pi(s_{t-1}, o) \Pr(T_{t-1} = 1 | H_{t-1} = h_{t-1}, \pi, \mathcal{O}) \big) \\
&+ \big( \Pr(O_{-1} = o | H_{t-1} = h_{t-1}, T_{t-1} = 0, \pi, \mathcal{O}) \\
&\times (1 - \Pr(T_{t-1} = 1 | H_{t-1} = h_{t-1}, \pi, \mathcal{O})) \big) \Big] \\
=& \Big[ \big( \pi(s_{t-1}, o) \beta_o(s_{t-1}) \big) + \\
&\times \big( \Pr(O_{-1} = o | H_{t-1} = h_{t-1}, T_{t-1} = 0, \pi, \mathcal{O})(1 - \beta_o(s_{t-1})) \big) \Big]
\end{aligned}
$$

Given that by convention, $\Pr(T_0 = 1 | H_0 = h_0, \pi, \mathcal{O}) = 1.0$, we are now left with figuring out how to calculate $\Pr(O_{-1} = o | H_{t-1} = h_{t-1}, T_{t-1} = 0, \pi, \mathcal{O})$

$$
\begin{aligned}
\Pr(O_{-1} = o | H_{t-1} = h_{t-1}, T_{t-1} = 0, \pi, \mathcal{O}) =& \Pr(O_{-2} = o, A_{t-2} = a_{t-2}, S_{t-1} = s_{t-1} | H_{t-1} = h_{t-1}, \pi, \mathcal{O}) \\
=& \Pr(A_{t-2} = a_{t-2}, S_{t-1} = s_{t-1} | O_{-2} = o, H_{t-1} = h_{t-1}, \pi, \mathcal{O}) \\
&\times \Pr(O_{-2} = o | H_{t-1} = h_{t-1}, \pi, \mathcal{O}) \\
=& \Pr(S_{t-1} = s_{t-1} | A_{t-2} = a_{t-2}, O_{-2} = o, H_{t-1} = h_{t-1}, \pi, \mathcal{O}) \\
&\times \Pr(A_{t-2} = a_{t-2} | O_{-2} = o, H_{t-1} = h_{t-1}, \pi, \mathcal{O}) \\
&\times \Pr(O_{-2} = o | H_{t-1} = h_{t-1}, \pi, \mathcal{O}) \\
=& P(s_{t-2}, a_{t-2}, s_{t-1}) \mu_o(s_{t-2}, a_{t-2}) \Pr(O_{-2} = o | H_{t-1} = h_{t-1}, \pi, \mathcal{O}) \\
=& P(s_{t-2}, a_{t-2}, s_{t-1}) \mu_o(s_{t-2}, a_{t-2}) \Pr(O_{-2} = o | H_{t-2} = h_{t-2}, \pi, \mathcal{O})
\end{aligned}
$$

where $\Pr(O_0 = o | H_0 = h_0, \pi, \mathcal{O}) = \pi(s_0, o)$

Using the recursive function $\Pr(O_{t-1} = o'|H_{t-1} = h_{t-1}, \pi, \mathcal{O})$, the expected number of terminations for a given trajectory is given by:

$$\sum_{t=1}^{|h|} \mathbf{E}\left[T_t = 1|H_t = h_t, \pi, \mathcal{O}\right] = \sum_{t=1}^{|h|} \left(\sum_{o\in\mathcal{O}} \beta_o(s_t) \frac{\mu_o(s_{t-1}, a_{t-1})\Pr(O_{t-1} = o|H_{t-1} = h_{t-1}, \pi, \mathcal{O})}{\sum_{o'\in\mathcal{O}} \mu_{o'}(s_{t-1}, a_{t-1})\Pr(O_{t-1} = o'|H_{t-1} = h_{t-1}, \pi, \mathcal{O})}\right),$$

$\square$

### A.3   Appendix C - Proof of Theorem 2

**Theorem 2**

*Given a set of options $\mathcal{O}$ and a policy $\pi$ over options, the probability of generating a trajectory $h$ of length $|h|$ is given by:*

$$\Pr(H_{|h|} = h_{|h|}|\pi, \mathcal{O}) = d_0(s_0)\left[\sum_{o\in\mathcal{O}} \pi(s_0, o)\mu_o(s_0, a_0)f(h_{|h|}, o, 1)\right]\prod_{k=0}^{|h|-1} P(s_k, a_k, s_{k+1}),$$

*where $f$ is a recursive function defined as:*

$$f(h_t, o, i) = \begin{cases} 1, & \text{if } i = t \\[2mm] \Big[\beta_o(s_i)\sum_{o'\in\mathcal{O}} \pi(s_{i+1}, o')\mu_{o'}(s_{i+1}, a_{i+1})f(h_t, o', i+1) \\[2mm] \quad +(1 - \beta_o(s_i))\mu_o(s_{i+1}, a_{i+1})f(h_t, o, i+1)\Big], & \text{otherwise} \end{cases}$$

*Proof.* We define $H_{i,t}$ to be the history from time $i$ to time $t$, that is, $H_{i,t} = (S_i, A_i, S_{i+1}, A_{i+1}, \ldots, S_t)$, where $i < t$. If $i = t$, the history would contain a single state.

$$\begin{aligned}
\Pr(H_t = h_t|\pi, \mathcal{O}) &= \Pr(S_0 = s_0|\pi, \mathcal{O})\Pr(H_{1,t} = h_{1,t}, A_0 = a_0|S_0 = s_0, \pi, \mathcal{O}) \\
&= d_0(s_0)\Pr(H_{1,t} = h_{1,t}, A_0 = a_0|S_0 = s_0, \pi, \mathcal{O}) \\
&= d_0(s_0)\sum_{o\in\mathcal{O}} \Pr(H_{1,t} = h_{1,t}, A_0 = a_0, O_o = o|S_0 = s_0, \pi, \mathcal{O}) \\
&= d_0(s_0)\sum_{o\in\mathcal{O}} \Pr(O_0 = o|S_0 = s_0, \pi, \mathcal{O})\Pr(H_{1,t} = h_{1,t}, A_0 = a_0|S_0 = s_0, O_0 = o, \pi, \mathcal{O}) \\
&= d_0(s_0)\sum_{o\in\mathcal{O}} \pi(s_0, o)\Pr(H_{1,t} = h_{1,t}, A_0 = a_0|S_0 = s_0, O_0 = o, \pi, \mathcal{O}) \\
&= d_0(s_0)\sum_{o\in\mathcal{O}} \pi(s_0, o)\Pr(A_0 = a_o|S_0 = s_0, O_0 = o, \pi, \mathcal{O}) \\
&\quad \times \Pr(H_{1,t} = h_{1,t}|S_0 = s_0, O_0 = o, A_0 = a_0, \pi, \mathcal{O}) \\
&= d_0(s_0)\sum_{o\in\mathcal{O}} \pi(s_0, o)\mu_o(s_0, a_o)\Pr(H_{1,t} = h_{1,t}|S_0 = s_0, O_0 = o, A_0 = a_0, \pi, \mathcal{O}).
\end{aligned}$$

We now need to find an expression to calculate $\Pr(H_{1,t} = h_{1,t}|S_0 = s_0, O_0 = o, A_0 = a_0, \pi, \mathcal{O})$. Consider the probability of seeing history $h_{i,t}$ given the previous state, $s$, the previous option, $o$, and the previous action, $a$:

$$\Pr(H_{i,t} = h_{i,t}|S_{i-1} = s, O_{i-1} = o, A_{i-1} = a)$$
$$= \Pr(S_i = s_i|S_{i-1} = s, O_{i-1} = o, A_{i-1} = a)\Pr(H_{i+1,t} = h_{i+1,t}, A_i = a_i|S_{i-1} = s, O_{i-1} = o, A_{i-1} = a, S_i = s_i)$$
$$= P(s, a, s_i)\Pr(H_{i+1,t} = h_{i+1,t}, A_i = a_i|S_{i-1} = s, O_{i-1} = o, A_{i-1} = a, S_i = s_i)$$
$$= P(s, a, s_i)\Pr(H_{i+1,t} = h_{i+1,t}, A_i = a_i|O_{i-1} = o, A_{i-1} = a, S_i = s_i)$$
$$= P(s, a, s_i)\big[\Pr(T_i = 1|O_{i-1} = o, A_{i-1} = a, S_i = s_i)$$
$$\times \Pr(H_{i+1,t} = h_{i+1,t}, A_i = a_i|O_{i-1} = o, A_{i-1} = a, S_i = s_i, T_i = 1)$$
$$+ \Pr(T_i = 0|O_{i-1} = o, A_{i-1} = a, S_i = s_i)$$
$$\times \Pr(H_{i+1,t} = h_{i+1,t}, A_i = a_i|O_{i-1} = o, A_{i-1} = a, S_i = s_i, T_i = 0)\big]$$
$$= P(s, a, s_i)\big[\beta_o(s_i)$$
$$\times \Pr(H_{i+1,t} = h_{i+1,t}, A_i = a_i|O_{i-1} = o, A_{i-1} = a, S_i = s_i, T_i = 1)$$
$$+ (1 - \beta_o(s_i))$$
$$\times \Pr(H_{i+1,t} = h_{i+1,t}, A_i = a_i|O_{i-1} = o, A_{i-1} = a, S_i = s_i, T_i = 0)\big].$$

Even though the equation above might seem complicated, there are only two cases we need to consider: either the current option terminates and a new one must be selected (the first term), or the current option does not terminate (the second term). Let's consider each of them separately.

**Case 1 - option terminates:** If we terminate, we sum over new options:

$$\Pr(H_{i+1,t} = h_{i+1,t}, A_i = a_i|O_{i-1} = o, A_{i-1} = a, S_i = s_i, T_i = 1)$$
$$= \sum_{o' \in \mathcal{O}} \Pr(O_i = o'|O_{i-1} = o, A_{i-1} = a, S_i = s_i, T_i = 1)$$
$$\times \Pr(H_{i+1,t} = h_{i+1,t}, A_i = a_i|O_{i-1} = o, A_{i-1} = a, S_i = s_i, T_i = 1, O_i = o')$$
$$= \sum_{o' \in \mathcal{O}} \pi(s_i, o')\Pr(H_{i+1,t} = h_{i+1,t}, A_i = a_i|O_{i-1} = o, A_{i-1} = a, S_i = s_i, T_i = 1, O_i = o')$$
$$= \sum_{o' \in \mathcal{O}} \pi(s_i, o')\Pr(H_{i+1,t} = h_{i+1,t}, A_i = a_i|S_i = s_i, O_i = o')$$
$$= \sum_{o' \in \mathcal{O}} \pi(s_i, o')\Pr(A_i = a_i|S_i = s_i, O_i = o')\Pr(H_{i+1,t} = h_{i+1,t}|S_i = s_i, O_i = o', A_i = a_i)$$
$$= \sum_{o' \in \mathcal{O}} \pi(s_i, o')\mu_{o'}(s_i, a_i)\Pr(H_{i+1,t} = h_{i+1,t}|S_i = s_i, O_i = o', A_i = a_i).$$

Note that the expanded probability has the same form as $\Pr(H_{i,t} = h_{i,t}|S_{i-1} = s, O_{i-1} = o, A_{i-1} = a)$.

**Case 2 - option does not terminate:** This tells us that $O_i = o$, so we may drop the dependency on the $i - 1$ terms:

$$\Pr(H_{i+1,t} = h_{i+1,t}, A_i = a_i|S_{i-1} = s, O_{i-1} = o, A_{i-1} = a, S_i = s_i, T_i = 0)$$
$$= \Pr(H_{i+1,t} = h_{i+1,t}, A_i = a_i|S_i = s_i, O_i = o)$$
$$= \Pr(A_i = a_i|S_i = s_i, O_i = 0)\Pr(H_{i+1,t} = h_{i+1,t}|S_i = s_i, O_i = o, A_i = a_i)$$
$$= \mu_o(s_i, a_i)\Pr(H_{i+1,t} = h_{i+1,t}|S_i = s_i, O_i = o, A_i = a_i).$$

Plugging these two cases back into our earlier equation yields:

$$\Pr(H_{i,t} = h_{i,t}|S_{i-1} = s, O_{i-1} = o, A_{i-1} = a)$$

$$=P(s, a, s_i)\Big[\beta_o(s_i) \sum_{o' \in \mathcal{O}} \pi(s_i, o')\mu_{o'}(s_i, a_i) \Pr(H_{i+1,t} = h_{i+1,t}|S_i = s_i, O_i = o', A_i = a_i)$$

$$+ (1 - \beta_o(s_i))\mu_o(s_i, a_i) \Pr(H_{i+1,t} = h_{i+1,t}|S_i = s_i, O_i = o, A_i = a_i)\Big].$$

Note that each term contains an expression of the same form, $\Pr(H_{i,t} = h_{i,t}|S_{i-1} = s, O_{i-1} = o, A_{i-1} = a)$. We can therefore compute the probability recursively. Our recursion will terminate when we consider $i = t$, as $H_{t,t}$ contains a single state, and we adopt the convention of its probability to be 1. Notice that for every recursive step, both inner terms will produce a $P(s, a, s_i)$ term. Consider the result when we factor every recursive $P(s, a, s_i)$ term to the front of the equation. We define the following recursive function:

$$f(h_t, o, i) = \begin{cases} 1, & \text{if } i = t \\ \Big[\beta_o(s_i) \sum_{o' \in \mathcal{O}} \pi(s_{i+1}, o')\mu_{o'}(s_{i+1}, a_{i+1})f(h_t, o', i+1) \\ \quad + (1 - \beta_o(s_i))\mu_o(s_{i+1}, a_{i+1})f(h_t, o, i+1)\Big], & \text{otherwise} \end{cases}$$

.

Notice that this is the recursive probability described above, but with the $P(s, a, s')$ terms factored out. We now see that:

$$\Pr(H_{i,t} = h_{i,t}|S_{i-1} = s_{i-1}, O_{i-1} = o, A_{i-1} = a_{i-1}) = f(h_t, o, i) \prod_{k=i-1}^{t-1} P(s_k, a_k, s_{k+1}).$$

Plugging this all the back into our original equation for $\Pr(H_t = h_t|\pi, \mathcal{O})$ gives us the desired result:

$$\Pr(H_{|h|} = h_{|h|}|\pi, \mathcal{O}) = d_0(s_0)\Big[\sum_{o \in \mathcal{O}} \pi(s_0, o)\mu_o(s_0, a_o)f(h_{|h|}, o, 1)\Big] \prod_{k=0}^{t-1} P(s_k, a_k, s_{k+1}).$$

$\square$

## A.4 APPENDIX D - EMPIRICAL VALIDATION OF DERIVED EQUATIONS

To double check the derivation of the proposed objective and make sure the implementation was correct, we conducted a simple empirical test to compared the calculated expected number of decisions in a trajectory and the probability of generating each trajectory for a set of 10 trajectories on 10 MDPs. The MDPs are simple chains of 7 states with different transition functions. We randomly initialized four options and a policy over options, and estimated the probability of generating each trajectory and the expected number of terminations, for each sampled trajectory, by Montecarlo sampling $10,000$ trials. Table 1 presents results for the 10 trajectories verifying empirically that the equations were correctly derived and implemented. The table compares the empirical and true probability of generating a given trajectory, $\hat{\Pr}(H|\cdot)$ and $\Pr(H|\cdot)$, respectively, and the empirical and true sum of expected number of decisions an agent has to make to generate those trajectories, $\sum_{t=1}^{|H|} \hat{\mathbf{E}}[T_t|\cdot]$ and $\sum_{t=1}^{|H|} \mathbf{E}[T_t|\cdot]$, respectively.

| H | $\hat{\Pr}(H|\pi,\mathcal{O})$ | $\Pr(H|\pi,\mathcal{O})$ | $\sum_{t=1}^{|H|}\hat{\mathbf{E}}\left[T_t|H_t,\pi,\mathcal{O}\right]$ | $\sum_{t=1}^{|H|}\mathbf{E}\left[T_t|H_t,\pi,\mathcal{O}\right]$ |
|---|---|---|---|---|
| $h_1$ | 0.0932 | 0.0957 | 3.060 | 3.178 |
| $h_2$ | 0.0158 | 0.0173 | 4.139 | 4.154 |
| $h_3$ | 0.2149 | 0.2122 | 1.965 | 2.178 |
| $h_4$ | 0.0995 | 0.0957 | 2.979 | 3.178 |
| $h_5$ | 0.0962 | 0.0957 | 3.024 | 3.178 |
| $h_6$ | 0.1354 | 0.1384 | 2.9579 | 3.1596 |
| $h_7$ | 0.00040 | 0.00038 | 9.750 | 8.794 |
| $h_8$ | 0.1854 | 0.1881 | 2.820 | 3.072 |
| $h_9$ | 0.0379 | 0.0368 | 4.2612 | 4.4790 |
| $h_{10}$ | 0.1864 | 0.1881 | 2.8404 | 3.0723 |

Table 1: Validation of equations and implementation.

Note that the cases with largest discrepancy between the estimated and calculated number of terminations occur when the probability of generating a trajectory is low. This happens because, since the trajectory is unlikely to be generated, the Monte Carlo sampling is not able to produce enough samples of the trajectory.

## A.5 APPENDIX E - IMPLEMENTATION DETAILS FOR ATARI EXPERIMENTS

For these experiments we first learned a good performing policy with A3C for each game and sampled 12 trajectories for training. Each trajectory lasted until a life was lost, not for the entire duration of the episode. Each option was represented as a two-layer neural network, with 32 neurons in each layer, and two output layers: a softmax output layer over the four possible actions representing $\mu$, and a separate sigmoid layer representing $\beta$. We implemented our objective using PyTorch which simplifies gradient calculations. The options were represented by a two-layer neural network, where the input was represented by gray scale images of the last two frames. We ran 32 training agents in parallel on CPUs, the learning rate was set to $0.0001$ and the discount factor $\gamma$ was set to $0.99$.

Because the options can only learn from the states observed in the trajectories, it is possible that when using them, they will be executed in previously unseen states. When this happens, the termination function may decide to never terminate, as it has not seen that region of the state space before. To address this issue, we add a value of $0.05$ to the predicted probability of termination per time-step that the option has been running since executed. Therefore, in our experiments an option cannot run for more than 20 time-steps in total.

