# OpenReview forum: "Learning Reusable Options for Multi-Task Reinforcement Learning"
_ICLR.cc/2020/Conference — Reject_

### Official Review · AnonReviewer1 · 2019-10-20
**Official Blind Review #1**

**Rating:** 3

**Review:**

The paper proposes a method for learning options that transfer across multiple learning tasks. The method takes a number of demonstration trajectories as input and attempts to create a set of options that can recreate the trajectories with minimal terminations.

I currently recommend rejection. The evaluation of the proposed method is rather weak and does not clearly demonstrate that the author’s goals have been achieved. The paper could also do a better job of situating the approach with regard to existing option learning approaches.

Detailed comments:

- The paper strongly emphasizes reusability of learnt options over multiple tasks as a key goal. This aspect is largely absent from the practical part of the paper, however. The proposed algorithm largely ignores the multi-task aspect beyond requiring demonstrations from different tasks - also see the next remark. In the experiments, the multi-task transfer is not emphasized. In the 4rooms domain options are learnt on training tasks and evaluated on test tasks, but the effect of task distribution or task diversity on generalisation is not investigated. Moreover, the larger scale ATARI experiments do not seem to include any multi-task aspects at all, with options immediately being learnt on the target task.

- The objective in (1) omits any explicit mention of different tasks. It would be good to indicate explicitly how it depends on the distribution of tasks and what the expectation is taken over.

- The authors indicate that their learning objective needs the transition function P for the MDP. This is never further discussed. Do the experiments assume known transition functions? If not, how are these functions estimated? If a model is known, does it still make sense to learn option policies from samples or would it be better to use  planning based options (see e.g.[1])?

- While the paper cites a number of option learning approaches, it could do a better job of situating the research within the literature. There are a number of option induction approaches that explicitly focus on reusability of options - see e.g. [1], [4]. There have also been a large number of approaches that focus on hierarchical learning to represent a distribution over demonstration trajectories: see eg. [3],[5], [6], [7]. Some of these approaches might also be better baselines than OptionCritic which doesn’t explicitly take into account learning from demonstrations or multi-task transfer.

[1] Mann, T. A., Mannor, S., & Precup, D. (2015). Approximate value iteration with temporally extended actions. JAIR, 53, 375-438.
[2] Konidaris, G., & Barto, A. G. (2007). Building Portable Options: Skill Transfer in Reinforcement Learning. In IJCAI,
[3] Konidaris, G., Kuindersma, S., Grupen, R., & Barto, A. (2012). Robot learning from demonstration by constructing skill trees. IJRR, 31(3), 360-375.
[4] Andreas, J., Klein, D., & Levine, S. (2017). Modular multitask reinforcement learning with policy sketches. ICML
[5] Henderson, P., Chang, W. D., Bacon, P. L., Meger, D., Pineau, J., & Precup, D. (2018). Optiongan: Learning joint reward-policy options using generative adversarial inverse reinforcement learning. AAAI.
[6] Co-Reyes, J. D., Liu, Y., Gupta, A., Eysenbach, B., Abbeel, P., & Levine, S. (2018). Self-consistent trajectory autoencoder: Hierarchical reinforcement learning with trajectory embeddings.ICML
[7] Daniel, C., Van Hoof, H., Peters, J., & Neumann, G. (2016). Probabilistic inference for determining options in reinforcement learning. Machine Learning, 104(2-3), 337-357.


Minor comments:
- The results on ATARI seem to have been ended before reaching final learning performance

- I couldn’t find details for how the transition function in 4-rooms is changed

- Does the optionCritic comparison include the deliberation cost? Since this paper aims to minimise option terminations that seems to be the most logical comparison.

- Why don’t the ATARI results compare against other approaches?

- The influence of the KL penalty isn’t really examined in results beyond looking at performance. How does it influence the trade-off between representing trajectories and diversity?


**Experience Assessment:**

I have published in this field for several years.

**Review Assessment: Checking Correctness Of Derivations And Theory:**

I assessed the sensibility of the derivations and theory.

**Review Assessment: Checking Correctness Of Experiments:**

I carefully checked the experiments.

**Review Assessment: Thoroughness In Paper Reading:**

I read the paper thoroughly.

---

> ### Author Response · Authors · 2019-11-07
> **Response to Reviewer #1**
>
> Thank you for taking the time to review our submission and for your feedback.
>
> - We are a bit puzzled on what you refer to by investigating "effect of task distribution or diversity"? The test tasks in the four room domain are varied by changing the goal location, which implies a change in the reward function. If you could clarify what you are refering to, we would gladly add a discussion about it.
>
> - We apologize for not making this point clearer. The experiments in the ATARI domain include a multi-task component; in each task we changed initial state (the agent was allowed to move for a random number of steps before beginning learning), the number of frames skipped per action taken, and magnitude of the reward.
>
> - In objective (1) the expectation is taken over trajectories which we assumed are sampled from a distribution over optimal policies for different tasks. We agree we could have made this more precise, and it's not explicitly reflected in the notation we used.
>
> - Thank you for the list of references. Taking your suggestion into consideration, we will try to expand our baselines in the allotted time.  We also agree we could have done a better job at describing these experimental details, and will make the necessary changes to the document to address these comments.
>
>
> ==============================
>
> I have updated the document to address some of your concerns:
>
> - Clarified how the multi-task setting was created for ATARI (Explained in experiment section)
> - Explained how the objective relies on multi tasks (Explained in problem formulation)
> - Clarified how we estimate the transition functions from samples while learning on training tasks (In Experiment section)
> - In four rooms we assumes that the transition function is known, in the ATARI experiments we assumed a linear-gaussian model and fit the parameters with maximum likelihood estimation.

---

> > ### Comment · AnonReviewer1 · 2019-11-14
> > **Follow up comments**
> >
> > Thank you for the explanations. These do clarify some of the issues I had with the paper.
> >
> > > what you refer to by investigating "effect of task distribution or diversity"
> >
> > My comment was mainly meant to indicate that I think the empirical evaluations are somewhat limited. You do not really vary the training and testing conditions. The main claim of the paper is that the method offers options that generalise to new tasks. The experiments show only learning curves for train on 1 set of tasks test on another set. There are many more things that could be evaluated to determine how robust the generalisation is depending on various conditions:
> >
> > - The number of tasks in the training set
> > - The number of samples generated on each training task
> > - How different the training tasks are (i.e. do they need to span the range of possible changes)
> > - How similar the distribution of training tasks and test tasks is
> > - What happens if the test set contains changes not seen in the training tasks?

---

### Official Review · AnonReviewer2 · 2019-10-22
**Official Blind Review #2**

**Rating:** 6

**Review:**

This paper proposes a new option discovery method for multi-task RL to reuse the option learned in previous tasks for better generalization. The authors utilize demonstrations collected beforehand and train an option learning framework offline by minimizing the expected number of terminations while encouraging diverse options by adding a regularization term. During the offline training, they add one option at a time and move onto the next option when the current loss fails to improve over the previous loss, which enables automatically learning the number of options without manually specifying it. Experiments are conducted on the four rooms environment and Atari 2600 games and demonstrate that the proposed method leads to faster learning on new tasks.

Overall, this paper gives a novel option learning framework that results in some improvement in multi-task learning. While the paper is technically sound and somewhat supported by experimental evidence, the experiments are limited to low-dimensional state space and discrete action space. I do wonder if the method can scale to high-dimensional space with continuous control.

Moreover, the framework requires optimal policies to generate trajectories for offline option learning, which seems to add more supervision signals than prior work such as option-critic. I wonder how the method would perform under sub-optimal demonstrations or even random trajectories generated by some RL policy.

Finally, I wonder how this method can be compared to skill embedding learning methods such as [1], which have been shown to be able to compactly represent skills in a latent space and reuse those skills in high-dimensional robotic manipulation tasks.

[1] Hausman, Karol, Jost Tobias Springenberg, Ziyu Wang, Nicolas Heess, and Martin Riedmiller. "Learning an embedding space for transferable robot skills." (2018).


**Experience Assessment:**

I have read many papers in this area.

**Review Assessment: Checking Correctness Of Derivations And Theory:**

I assessed the sensibility of the derivations and theory.

**Review Assessment: Checking Correctness Of Experiments:**

I assessed the sensibility of the experiments.

**Review Assessment: Thoroughness In Paper Reading:**

I read the paper at least twice and used my best judgement in assessing the paper.

---

> ### Author Response · Authors · 2019-11-07
> **Response to Reviewer #2**
>
> Thank you for taking the time to review our submission; we hope the following addresses some of your concerns.
>
> - It is true that in our experiments we have focused on discrete action space problems, but, in the case of atari, the state space is high dimensional. The input to the options is given as a stack of the last two frames of the game; in our opinion that is not necessarily low dimensional.
> In theory, continuous control can be modeled as well, but one would have to assume a family of distribution for the option policy to optimize. For example, one could assume that the option follows a gaussian distribution, and to maximize the probability of generating the observed trajectories one would optimize the mean and variance.
>
> - We apologize for the confusion, maybe we could have phrased this better. The method does not require optimal trajectories, but that is an assumption we make (or at least they are generated by a policy that performs well). The reason for this is that if the trajectories are obtained from a poorly performing policy (for example, random), the options learned will not be useful. They will fit that poor performing pattern.
> We will update the paper to make this more clear.
>
>
> - Thank you for pointing out the work in [1], that is definitely a paper worth mentioning. However, their setting is a bit different from ours. The method [1] requires to have a task ID (one hot encoding) to learn the embeddings. That means that the number of possible different tasks must be known apriori. In our case, we only require sample trajectories to learn options that can be used across many tasks.
>
>
>
>
> ==============================
>
> I have updated the document to address some of your concerns:
>
> - Clarified why we assume that the trajectories are obtained from optimal policies (Explained in section 3, before problem formulation).

---

### Official Review · AnonReviewer3 · 2019-10-23
**Official Blind Review #3**

**Rating:** 1

**Review:**

Summary:
The authors propose to learn reusable options to make use of prior information and claim to do so with minimal information from the user (such as # of options needed to solve the task, which options etc). The claim is that the agent is first able to learn a near-optimal policy for a small # of problems and then is able to solve a large # of tasks by such a learned policy.  The authors build on the idea that minimizing the number of decisions made by the agent results in discovering reusable options. The options are learned offline by learning to solve a small number of tasks. Their algorithm introduces one option at a time until introducing a new option doesn’t improve the objective further. The ideas are interesting, However, the paper as it stands is lacking in thorough evaluation.

Detailed comments:
The proposed approach offers two key contributions:
-an objective function to minimize the decision states
-incrementally constructing an option set that can be reused later, without the a priori specification of the # of options needed.

The introduction is well written, however, given the intuitions behind the objective function; in some sense, the idea here is to minimize the decisions or terminations intuitively relates to terminating only at critical or bottleneck states. It would be useful to provide such motivation in the introduction.

Intuitively the objective criterion is interesting. With a cursory look at the proofs, they seem fine, although I have to admit I have not looked in detail into the proofs.

Paper writing could be significantly improved. Several points are not clear and need further clarification:
-The term near-optimal is mentioned several times, but it is not clear the policies are near-optimal with respect to what? The task or a set of tasks?
-How does the proposed approach ensures that they are near-optimal? Please clarify.
-“We can obtain the estimate for equation 1 by averaging over a set of near-optimal trajectories” The aim as states is to learn options that are capable of generating near-optimal trajectories (by using a small # of terminations). The authors then say that “given a set of options, a policy over options, a near-optimal sample trajectory, we can calculate..” Where does the near-optimal sample trajectory come from? Please provide clarifications.

In experiments: FR rooms experiments are interesting, in the visualization of the option policies, do the figures here show the flattened policy of the options? What do the termination functions look like?

Atari experiments are limited in nature in that they show only two games. Moreover, It is a bit confusing as to what is multi-task in the ATARI experiments. The authors mention the training of options and then talk about the results in the plots (4) show the training curves. However, they do not mention what are “novel tasks for Breakout/Amidar” in this context.

Considering the proposed approach is closely related to the idea of selective terminations of options, it is natural to expect a comparison with Harb, 2018 and Hartyuanm 2019. The work could benefit by comparing with the aforementioned baselines. In particular, the visualization in 4b showing options learned in Amidar does not show much improvement from what was observed before in Harb, 2018.

With the motivation of this paper, I am unable to convince myself about options being “reusable” for multi-task here. It would be very useful for the reader to clarify what “novel tasks” are here to appreciate what is learned. Looking deeper into the appendix, I understand that the authors “first learned a good performing policy with A3C for each game and sample 12 trajectories for training.” This is not at all clear in the main paper. Besides, what does it mean by a "good" policy? If we already have that, it is unclear what gains do we get from the proposed method.

One obvious limitation here is that they also have a hard imposed constraint here is that the options cannot run for more than 20 time-steps in total, to make the objective function a suitable choice.

Overall:
An interesting objective function, Learn not only option set but also the number of options needed and incrementally learn new options.

Paper writing does not convey clearly what are novel tasks and could be significantly improved.

Since the paper claims multi-task and mentions several lifelong learning works like [1], I was expecting rigorous baselines showing performance over multiple tasks. The experiments are lacking in that evidence except for four rooms domain, which is much simpler a domain.

Near-optimal property is very much lacking the clarity to the best of my knowledge.

[1]Ammar, Haitham Bou, et al. "Online multi-task learning for policy gradient methods." International Conference on Machine Learning. 2014.

**Experience Assessment:**

I have published one or two papers in this area.

**Review Assessment: Checking Correctness Of Derivations And Theory:**

I assessed the sensibility of the derivations and theory.

**Review Assessment: Checking Correctness Of Experiments:**

I carefully checked the experiments.

**Review Assessment: Thoroughness In Paper Reading:**

I read the paper thoroughly.

---

> ### Author Response · Authors · 2019-11-07
> **Response to Reviewer #3**
>
> We appreciate the time you took reviewing our submission and hope the our response help address some of your concerns.
>
> - It might seem intuitive that options would terminate at bottleneck states, but that in our formulation there is no explicit notion of bottleneck states, and the options do not have to terminate at those states. Notice that we are not only minimizing the number of terminations, but jointly maximizing the probability of generating the given trajectories. So our objective might learn that options have to cross bottleneck states, but terminate after crossing them.
>
> - We apologize for the vagueness in 'near-optimal'. What we meant by this is that the trajectories are generated by a learned (well-performing policy), but they might not be technically "optimal".
>
> - The proposed approach does not ensure they are near-optimal, it assumes they are. The agent learns a policy in a few training tasks, and we assume that these policies perform well.
>
> - The sample trajectories are generated when learning on the training tasks. As the agent is learning a policy, it is generating trajectories from each updated policy. We used the trajectories generated by the policy learned at the end of the training process.
>
> - In the experiments for FR, the flattened policy is shown in the top right figured labeled "Solution Found". The remaining figures show the option policies, before and after learning, for each respective option.
> For the termination functions, it is a bit difficult to visualize so many dimensions (option policy, policy over options, how the options are used, etc...). We will do our best to update the document depicting the termination functions as well.
>
> - Reviewer #1 also mentioned the multi-task setting in ATARI, and we apologize for not making this clear. We reiterate the response here:
> The experiments in the ATARI domain include a multi-task component; in each task we changed initial state (the agent was allowed to move for a random number of steps before beginning learning), the number of frames skipped per action taken, and magnitude of the reward.
>
> - In response to your comment: "If we already have that, it is unclear what gains do we get from the proposed method". The point we are making is that if you have a policy that works well for the task you are currently addressing, it is likely that it won't work well for a new task. A change in the transition function or reward function will probably change how that policy performs.  However, there is useful information we can leverage from those tasks that are different, but related to the new task. We are showing that one way of doing that is to learn options that we can reuse for any novel, but related, task the agent might face.
>
> - We will update our submission to clarify many of the points we could have explained better.
>
>
> ==============================
>
> I have updated the document to address some of your concerns:
>
> - Added a discussion on how our method differs from finding bottleneck states. ( Explained in introduction last paragraph)
> - Clarify what we mean by (near)-optimal policies and why we use that term (Explained in last pragraph of introduction).
> - Updates the figures in four-rroms to include a visualization of termination functions.
>
> - To comment on the difference in performance from what was reported in Harb 2018. Keep in mind that by creating task variations, we are changing what return can be achieve at any particular task.
> - We clarified how ATARI experiments are mult-task (explained in experiments section).

---

> > ### Comment · AnonReviewer3 · 2019-11-14
> > **Follow up questions**
> >
> > > The proposed approach does not ensure they are near-optimal, it assumes they are.
> > Unfortunately, it is not clear how does this assumption hold still. Is it right to say by near-optimal , the authors mean a good policy and has nothing to do with optimal policy $\pi^{*}$ technically?
> >
> > > The experiments in the ATARI domain include a multi-task component; in each task we changed initial state..
> > How are the train and test start state data generated? How is the generalization being measured?
> >
> > > The experiments lack thoroughness and rigor in evaluations: the ATARI results do not compare against other baselines and make it hard to follow any claims made by the authors.
> >
> > > Besides, the authors show only 2 games in Atari: How were these 2 games chosen?
> >
> > > Would the multi-task learning extend across different games similar in the nature of the games?
> >
> > > I am concerned about the hard imposed constraint that the options cannot run for more than 20 time-steps in total. How could we lift this constraint in the context of the proposed approach? I believe it is may be not possible to lift this constraint, as otherwise the objective function would not as proposed would not be a suitable choice.

---

> > > ### Author Response · Authors · 2019-11-14
> > > **Follow up comments**
> > >
> > > Thank you for the suggestions and following up with the review.
> > >
> > > - Yes, it is right to say that by near-optimal we mean a good policy. We mean to say the best performing policy that was learned for the training tasks. Ideally it would be optimal, but we have no way of guaranteeing that it truly is optimal.
> > >
> > > - We clarified in the updated document how the tasks for ATARI are being generated. In short, we did the following: 1) draw a number between 2 and 10 to define how many frames are skipped after taking an action (changes transition function), 2)  draw a number between 1 and 20 and let the agent execute that many actions before starting learning (changes the initial state distribution), 3) scaling the reward by a number between 0.1 and 10 (changes the reward function).
> > >
> > > - The claim that we are making is that the proposed objective is able to learn options that generalize well across several new tasks, provided there are some similarities that can be exploited.  We are not saying that this will necessarily outperform other learning algorithms, but rather that we can learn options that can be added to the action set and will help the agent in learning.
> > >
> > > - We chose these two games because there are intuitive skills that might be learned (such as the paddle in breakout moving in one direction for a number of steps to catch a bouncing ball). We thought that being able to show that we can learn some of these behaviors would make for a compelling argument.
> > >
> > > - For this approach would extend across games similar in nature, it would depend on how states and actions are represented.  Our method does not take into account where the trajectories are coming from, it just seeks to learn options to recreate those trajectories.
> > > If there is an actions set and state space that is similar enough across games, then it should generalize in those cases.
> > > However, if you are dealing with a different state space, then I wouldn't expect our method to learn good options.
> > >
> > > - I understand your concern on lifting this contrained on the duration of an option. The difficulty is that when executing an option in a state that was never seen in a demonstration, the termination function never had a chance to learn what to do. So it might choose to not terminate.
> > > One way to lift this constraint might be for the termination function to also be able to update online (as the agent is learning), however, this is not a trivial change.

---

### Decision · Program_Chairs · 2019-12-19

**Decision:**

Reject

**Comment:**

This paper presents a novel option discovery mechanism through incrementally learning reusasble options from a small number of policies that are usable across multiple tasks.

The primary concern with this paper was with a number of issues around the experiments. Specifically, the reviewers took issue with the definition of novel tasks in the Atari context. A more robust discussion and analysis around what tasks are considered novel would be useful. Comparisons to other option discovery papers on the Atari domains is also required.

Additionally, one reviewer had concerns on the hard limit of option execution length which remain unresolved following the discussion.

While this is really promising work, it is not ready to be accepted at this stage.